# AN EFFICIENT HOMOTOPY TRAINING ALGORITHM FOR NEURAL NETWORKS

## ABSTRACT

We present a Homotopy Training Algorithm (HTA) to solve optimization problems arising from neural networks. The HTA starts with several decoupled systems with low dimensional structure and tracks the solution to the high dimensional coupled system. The decoupled systems are easy to solve due to the low dimensionality but can be connected to the original system via a continuous homotopy path guided by the HTA. We have proved the convergence of HTA for the non-convex case and existence of the homotopy solution path for the convex case. The HTA has provided a better accuracy on several examples including VGG models on CIFAR-10. Moreover, the HTA would be combined with the dropout technique to provide an alternative way to train the neural networks.

## 1 INTRODUCTION

Deep learning methods, as the rising star among all machine learning methods in recent years, have already had great success in many applications. Many advancements Cang et al. (2018); Cang & Wei (2018); Yin et al. (2018; 2016) on deep learning have been made in the last few years. However, as the size of new state-of-the-art models keep growing larger and larger, they rely more and more heavily on efficient algorithms for training and making inferences from such models. This clearly puts strong limitations on the application scenarios of nerual netowrks on roboticsKonda et al. (2012), auto-pilot automobiles Hammerla et al. (2016), and aerial systems Maire et al. (2014). At present, there are two big challenges in fundamentally understanding deep neural networks:

- How to efficiently solve the highly nonlinear and non-convex optimization problems that arise during the training of a deep learning model?
- How to design a deep neural network structure for specific problems?

In order to solve these challenges, in this paper, we will present a new training algorithm based on homotopy continuation method Bates et al. (2011; 2008); Morgan & Sommese (1987) which has been successfully used to study nonlinear problems such as nonlinear differential equations and optimizations Hao et al. (2012); Hao & Harlim. In order to tackle the nonlinear optimization problem in neural networks, the homotopy training algorithm is designed and shows efficiency and feasibility for both convolutional and fully connected neural networks with complex structures. The idea of the HTA is to start with a simplified optimization problem and construct a continuous path to the original problem. Specifically, let's consider the following general optimization problem

$$\min_{\theta} f(\theta) \text{ where } \theta \in \mathbb{R}^n. \tag{1}$$

We then divide $\theta$ in two groups, namely, $\theta = \theta_1 \cup \theta_2$, where $\theta_1 \in \mathbb{R}^{n_1}$ and $\theta_2 \in \mathbb{R}^{n_2}$ $(n_1 + n_2 = n)$. (Although that it can be easily extended to any $m$ decompositions, we use $m = 2$ as an illustration of the HTA idea.) Then we first solve the following decoupled simplified optimization problems

$$\min_{\theta_1} f(\theta_1, \tilde{\theta}_2) \text{ and } \min_{\theta_2} f(\tilde{\theta}_1, \theta_2) \tag{2}$$

where $\tilde{\theta} = \tilde{\theta}_1 \cup \tilde{\theta}_2$ is a random chosen point in $\mathbb{R}^n$ for instance we choose $\tilde{\theta} = 0$ in the dropout Srivastava et al. (2014). Second we define the following homotopy function

$$H(\Theta, L) = f(\theta) + L\|\theta_1 - \tilde{\theta}_1\|^2 + f(\eta) + L\|\eta_2 - \tilde{\theta}_2\|^2 + \frac{1}{L}\|\theta - \eta\|^2, \tag{3}$$

where $\Theta = (\theta, \eta)$ and $L \in (0, \infty)$ is the homotopy parameter. When $L \to \infty$, we have two decoupled optimization problems which we have already solved in the first step; when $L \to 0$, we recover the original optimization by forcing $\theta = \eta$. By gradually increasing parameter $L$, we are able to construct a continuous path $\Theta(L)$ between two decoupled low-dimensional optimization problems and the original optimization problem.

**Remark** The HTA can be reviewed in point view of nonlinear equations by considering the necessary condition of the optimization problem (1):

$$
H(\Theta, L) = \begin{pmatrix} \nabla_\theta f(\theta) \\ \eta - \theta \end{pmatrix} \frac{1}{L} + L \begin{pmatrix} \nabla_{\theta_2} f(\theta) \\ \theta_1 - \tilde{\theta}_1 \\ \nabla_{\eta_1} f(\eta) \\ \eta_2 - \tilde{\theta}_2 \end{pmatrix}.
$$

When $L$ is large, $\theta_2$ and $\eta_1$ can be solved by $\nabla_{\theta_2} f(\theta) = 0$ and $\nabla_{\eta_1} f(\eta) = 0$ respectively. When $L$ is small, we have the solution of $\nabla_\theta f(\theta) = 0$ by the diagonal homotopy $\theta - \eta = 0$. This technique has been widely used in solving systems of nonlinear equations especially for polynomial systems Bates et al. (2008); Wampler & Sommese (2005). Since the $\tilde{\theta}$ is randomly chosen, the universal approximation theory Park & Sandberg (1991) can guarantee that the random point $\tilde{\theta}$ in $\mathbb{R}^n$ is always connected to the solution of $\nabla_\theta f(\theta) = 0$. In another word, the solution path $\Theta(L)$ always exists or the probability that $\Theta(L)$ is not connected is zero Wampler & Sommese (2005).

## 2 APPLICATIONS OF HTA ON NEURAL NETWORK

### 2.1 CONVOLUTIONAL NEURAL NETWORKS

In this subsection, we will specify the variable $\theta$ and the objective function $f(\theta)$ of (3) in the convolutional neural network (CNN) setup.

**Definition 2.1.** *Suppose we have two tensors, $A_{l \times m \times n} = (a_{i,j,k})_{1 \leq i \leq l, 1 \leq j \leq m, 1 \leq k \leq n}$, $B_{c \times l \times m' \times n'} = (b_{h,i,j,k})_{1 \leq h \leq c, 1 \leq i \leq l, 1 \leq j \leq m', 1 \leq k \leq n'}$, where $c, m, n, m', n', l \in \mathbb{N}$ and $m' \leq m, n' \leq n$. The convolution of two tensors is defined by the following:*

$$
A * B_{h,j,k} = \sum_{i'=1}^{l} \sum_{j'=1}^{m'} \sum_{k'=1}^{n'} a_{i',j'+j-1,k'+k-1} \cdot b_{h,i',j',k'}, \tag{4}
$$

*where $B$ is called the convolutional kernel.*

Then with the input $x$ and the output $y$, the CNN is written as

$$
y(x; \theta) = \sigma(\sigma(\cdots \sigma(\sigma(x * W_1 + b_1) * W_2 + b_2) \cdots + b_{n-1}) * W_n + b_n) \tag{5}
$$

where $W_i \in \mathbb{R}^{c_i, l_i, m_i, n_i}$ and $\sigma$ is the activation function. Then we have $\theta = \{W_i, b_i\}_{i=1}^n$ which can be easily divided into two subgroups $\theta_j = \{W_{ij}, b_i\}_{i=1}^n$ where $W_{ij} \in \mathbb{R}^{\frac{c_i}{2}, l_i, m_i, n_i}$. The objective function $f(\theta)$ in (3) is defined by the cross entropy loss function Goodfellow et al. (2016) and defined as $\mathcal{L}(x, y) = -x(y) + log(\sum_j e^{x[j]})$. Due to the random algorithms in machine learning Bottou (2010); Kingma & Ba (2014), the homotopy function in (3) can be rewritten as $H(\Theta; \xi, L)$ where $\xi$ is a random variable due to random algorithms.

### 2.2 FULLY CONNECTED NEURAL NETWORKS

Similar to the CNN, a fully connected neural network with $n$ hidden layers is written as

$$
y(x, \theta) = \sigma(W_n \sigma(W_{n-1} \sigma(W_2 \sigma(W_1 x + b_1) + b_2) \cdots + b_{n-1}) + b_n), \tag{6}
$$

where $W_i \in R^{n \times d_i}$, $\beta_i \in R^{d_i}$, $n$ is the dimension of the input $x$ and $d_i$ is the width of the i-th layer. Then we have $\theta = \{W_i, b_i\}_{i=1}^n$ which can be easily divided into two subgroups $\theta_1 = \{W_i, b_i\}_{i=1}^{n/2}$ and $\theta_2 = \{W_i, b_i\}_{i=n/2+1}^n$. Or we can use the dropout technique to randomly split $\theta$ as $\theta_1$ and $\theta_2$. Similarly, the homotopy function in (3) can be also rewritten as $H(\Theta; \xi, L)$ due to the random algorithms Bottou (2010); Kingma & Ba (2014).

## 3 Theoretical Analysis

In this section, we will provide a theoretical analysis of the HTA by using stochastic gradient scheme to solve (3) for each $L$. We redefine our stochastic gradient scheme for the homotopy process as

$$\theta_{k+1} = \theta_k - \gamma_k G(\theta_k; \xi_k, L_k), \tag{7}$$

where $G(\theta; \xi, L) := \nabla_\theta F(\theta; \xi, L)$, $\gamma_k$ is the learning rate, $L_0 = M$, $L_k \nearrow 0$ and $M$ is a given large number. Then we consider the expectation of the homotopy function as $\mathbb{E}_\xi(H(\theta; \xi, L))$, for simplicity, and denote $\mathbb{E}_\xi[H(\theta; \xi, L)]$ as $g(\theta; L)$.

### 3.1 Convergence of the HTA for any given $L$ in the non-convex case

**Theorem 3.1. (Nonconvex Convergence)** *If $\{\theta_k\}$ is contained in a bounded open set, supposing that (7) is run with a step size sequence satisfying*

$$\sum_{k=1}^{\infty} \gamma_k = \infty \ and \ \sum_{k=1}^{\infty} \gamma_k^2 < \infty, \tag{8}$$

*then we have*

$$\mathbb{E}[\frac{1}{A_k} \sum_{k=1}^{K} \gamma_k \|\nabla g(\theta_k; L)\|_2^2] \to 0 \ as \ K \to \infty \ \ with \ A_k := \sum_{k=1}^{K} \gamma_k. \tag{9}$$

*Proof.* First, we prove the **Lipschitz-continuous objective gradients** condition Bottou et al. (2018), which means that $g(\theta; L)$ is $C^1$ and $\nabla g(\theta; L)$ is Lipschitz continuous:

- $g(\theta; L)$ **is $C^1$.** Since $y(x_\xi; \theta) \in C^1$ and $\mathcal{L}(\cdot, y)$ is $C^1$, we have that $H(\theta; \xi, L) \in C^1$ or that $\nabla_\theta H(\theta; \xi, L)$ is continuous. Considering

$$\nabla_\theta g(\theta; L) = \nabla_\theta \mathbb{E}_\xi(H(\theta; \xi, L)) = \mathbb{E}_\xi(\nabla_\theta H(\theta; \xi, L)), \tag{10}$$

  we have that $\nabla_\theta g(\theta, L)$ is continuous or that $g(\theta; L) \in C^1$.

- $\nabla g(\theta; L)$ **is Lipschitz continuous.** Since $\nabla_\theta g(\theta; L) = \mathbb{E}_\xi(\nabla_\theta H(\theta; \xi, L))$, we only need to prove $\mathbb{E}_\xi(\nabla_\theta \mathcal{L}(y(x_\xi; \theta), y_\xi))$ is Lipschitz continuous. In fact

$$\nabla_\theta \mathcal{L}(y(x_\xi; \theta), y_\xi) = \nabla_x \mathcal{L}(y(x_\xi; \theta), y_\xi) \cdot \nabla_\theta y(x_\xi; \theta) \tag{11}$$

  we will show that $\nabla_x \mathcal{L}(y(x_\xi; \theta), y_\xi)$ and $\nabla_\theta y(x_\xi; \theta)$ are bounded and Lipschitz continuous. Since that both $\sigma(x) = \frac{1}{1+e^{-x}}$ and $\sigma'(x) = \sigma(x)(1-\sigma(x))$ are bounded and Lipschitz continuous, and that $\{\theta_k\}$ is in an open set (assumption of Theorem 3.1), $\nabla_\theta y(x_\xi; \theta)$ is bounded and Lipschitz continuous. ($x_\xi$ is bounded because the size of our dataset is finite.) By differentiating $\mathcal{L}(x, y)$, we have

$$\nabla_x \mathcal{L}(x, y) = (-\delta_y^1 + \frac{e^{x_1}}{\sum_j e^{x_j}}, \cdots, -\delta_y^n + \frac{e^{x_n}}{\sum_j e^{x_j}}), \tag{12}$$

  where $\delta$ is the Kronecker delta. since

$$\frac{\partial}{\partial x_k} \frac{e^{x_i}}{\sum_j e^{x_j}} = \begin{cases} \frac{e^{x_i} \sum_j e^{x_j} - (e^{x_i})^2}{(\sum_j e^{x_j})^2} & k = i \\ \frac{-e^{x_i} e^{x_k}}{(\sum_j e^{x_j})^2} & k \neq i, \end{cases} \tag{13}$$

  which implies that $\left| \frac{\partial}{\partial x_k} \frac{e^{x_i}}{\sum_j e^{x_j}} \right| \leq 2$, we have $\nabla_x \mathcal{L}(\cdot, y)$ is Lipschitz continuous and bounded. Therefore $\nabla_x \mathcal{L}(y(x_\xi; \theta), y_\xi)$ is Lipschitz continuous and bounded. Therefore we have that $\nabla g(\theta; L)$ is Lipschitz continuous.

Second, we prove the **first and second moment limits** condition Bottou et al. (2018):

a. According to our theorem assumption, $\{\theta_k\}$ is contained in an open set which is bounded. Since that $g$ is continuous, $g$ is bounded;

b. Since that $G(\theta_k; \xi_k, L) = \nabla_\theta F(\theta_k; \xi_k, L)$ is continuous, we have

$$\mathbb{E}_{\xi_k}[G(\theta_k; \xi_k, L)] = \nabla_\theta \mathbb{E}_{\xi_k}[H(\theta_k; \xi_k, L)] = \nabla_\theta g(\theta_k; L). \tag{14}$$

Therefore

$$\nabla g(\theta_k; L)^T \mathbb{E}_{\xi_k}[G(\theta_k; \xi_k, L)] = \nabla g(\theta_k; L)^T \cdot \nabla g(\theta_k; L) = \|\nabla g(\theta_k; L)\|_2^2 \geq u \|\nabla g(\theta_k; L)\|_2^2 \tag{15}$$

for $0 < u \leq 1$. On the other hand, we have

$$\|\mathbb{E}_{\xi_k}[G(\theta_k; \xi_k, L)]\|_2 = \|\nabla g(\theta_k; L)\|_2 \leq u_G \|\nabla g(\theta_k; L)\|_2^2 \tag{16}$$

for $u_G \geq 1$.

c. Since that $\nabla H(\theta_k; \xi_k, L)$ is bounded for a given $L$, we have $\mathbb{E}_{\xi_k}[\|\nabla H(\theta_k; \xi_k, L)\|_2^2]$ is also bounded. Thus

$$\mathbb{V}_{\xi_k}[G(\theta_k; \xi_k, L)] := \mathbb{E}_{\xi_k}[\|G(\theta_k; \xi_k, L)\|_2^2] - \|\mathbb{E}_{\xi_k}[G(\theta_k; \xi_k, L)]\|_2^2 \leq \mathbb{E}_{\xi_k}[\|G(\theta_k; \xi_k, L)\|_2^2], \tag{17}$$

which implies that $\mathbb{V}_{\xi_k}[G(\theta_k; \xi_k, L)]$ is bounded.

We have checked assumptions 4.1 and 4.3 in Bottou et al. (2018). By theorem 4.10 in Bottou et al. (2018), with the diminishing stepsize, namely,

$$\sum_{k=1}^{\infty} \gamma_k = \infty \text{ and } \sum_{k=1}^{\infty} \gamma_k^2 < \infty, \tag{18}$$

the following convergence is obtained

$$\mathbb{E}[\frac{1}{A_k} \sum_{k=1}^{K} \gamma_k \|\nabla g(\theta_k; L)\|_2^2] \to 0 \text{ as } K \to \infty. \tag{19}$$

$\square$

## 3.2 EXISTENCE OF SOLUTION PATH $\theta(L)$ FOR THE CONVEX CASE

Second we explore the existence of solution path $\theta(L)$ when $t$ varies from $M$ to 0 for the convex case theoretically. This is quite difficult for the non-convex case although we may find it numerically. However, theoretical analysis of the convex case would also give us some guidance on how the HTA obtains an optimal solution for $L = 0$.

**Theorem 3.2.** *Assume that $g(\cdot, \cdot)$ is convex and differentiable and that $\|G(\theta; \xi, L)\| \leq M_0$. Then for stochastic gradient scheme (7), we have*

$$\lim_{n \to \infty} \mathbb{E}[g(\bar\theta_n, \bar L_n)] = g(\theta_*^1, 0), \tag{20}$$

*where $\bar\theta_n = \frac{\sum_{k=0}^n \gamma_k \theta_k}{\sum_{k=0}^n \gamma_k}$ and $\bar L_n = \frac{\sum_{k=0}^n \gamma_k L_k}{\sum_{k=0}^n \gamma_k}$.*

*Proof.*

$$\begin{aligned}
\mathbb{E}[\|\theta_{k+1} - \theta_*^{L_{k+1}}\|^2] &= \mathbb{E}[\|\theta_k - \gamma_k G(\theta_k; \xi_k, L_k) - \theta_*^{L_k}\|^2] \\
&\quad - 2\mathbb{E}[\langle \theta_k - \gamma_k G(\theta_k; \xi_k, L_k) - \theta_*^{L_k}, \theta_*^{L_{k+1}} - \theta_*^{L_k}\rangle] \\
&\quad + \mathbb{E}[\|\theta_*^{L_{k+1}} - \theta_*^{L_k}\|^2].
\end{aligned} \tag{21}$$

By defining

$$A_k = -2\mathbb{E}[\langle \theta_k - \gamma_k G(\theta_k; \xi_k, L_k) - \theta_*^{L_k}, \theta_*^{L_{k+1}} - \theta_*^{L_k}\rangle] + \mathbb{E}[\|\theta_*^{L_{k+1}} - \theta_*^{L_k}\|^2], \tag{22}$$

and assuming that $\sum_{k=0}^{m} A_k \leq A < \infty$, we obtain

$$
\begin{aligned}
\mathbb{E}[\|\theta_{k+1} - \theta_*^{L_{k+1}}\|^2] &= \mathbb{E}[\|\theta_k - \gamma_k G(\theta_k; \xi_k, L_k) - \theta_*^{L_k}\|^2] + A_k \\
&= \mathbb{E}[\|\theta_k - \theta_*^{L_k}\|^2] - 2\gamma_k \mathbb{E}[\langle G(\theta_k; \xi_k, L_k), \theta_k - \theta_*^{L_k}\rangle] + \gamma_k^2 \mathbb{E}[\|G(\theta_k; \xi_k, L_k)\|^2] + A_k \\
&\leq \mathbb{E}[\|\theta_k - \theta_*^{L_k}\|^2] - 2\gamma_k \mathbb{E}[\langle G(\theta_k; \xi_k, L_k), \theta_k - \theta_*^{L_k}\rangle] + \gamma_k^2 M^2 + A_k
\end{aligned}
$$

Since that

$$
\begin{aligned}
\mathbb{E}[\langle G(\theta_k; \xi_k, L_k), \theta_k - \theta_*^{L_k}\rangle] &= \mathbb{E}_{\xi_0, \cdots, \xi_{k-1}}[\mathbb{E}_{\xi_k}[\langle G(\theta_k; \xi_k, L_k), \theta_k - \theta_*^{L_k}\rangle | \xi_0, \cdots, \xi_{k-1}]] \\
&= \mathbb{E}_{\xi_0, \cdots, \xi_{k-1}}[\langle \nabla g(\theta_k; L_k), \theta_k - \theta_*^{L_k}\rangle | \xi_0, \cdots, \xi_{k-1}] \\
&= \mathbb{E}[\langle \nabla g(\theta_k; L_k), \theta_k - \theta_*^{L_k}\rangle],
\end{aligned}
\tag{23}
$$

we have

$$
\mathbb{E}[\|\theta_{k+1} - \theta_*^{L_{k+1}}\|^2] \leq \mathbb{E}[\|\theta_k - \theta_*^{L_k}\|^2] - 2\gamma_k \mathbb{E}[\langle \nabla g(\theta_k; L_k), \theta_k - \theta_*^{L_k}\rangle] + \gamma_k^2 M^2 + A_k. \tag{24}
$$

Due to the convexity of $g(\cdot, L_k)$, namely,

$$
\langle \nabla g(\theta_k, L_k), \theta_k - \theta_*^{L_k}\rangle \geq g(\theta_k; L_k) - g(\theta_*^{L_k}; L_k), \tag{25}
$$

we conclude that

$$
\mathbb{E}[\|\theta_{k+1} - \theta_*^{L_{k+1}}\|^2] \leq \mathbb{E}[\|\theta_k - \theta_*^{L_k}\|^2] - 2\gamma_k \mathbb{E}[g(\theta_k; L_k) - g(\theta_*^{L_k}, L_k)] + \gamma_k^2 M^2 + A_k, \tag{26}
$$

or

$$
2\gamma_k \mathbb{E}[g(\theta_k; L_k) - g(\theta_*^{L_k}; L_k)] \leq -\mathbb{E}[\|\theta_{k+1} - \theta_*^{L_{k+1}}\|^2 - \|\theta_k - \theta_*^{L_k}\|^2] + \gamma_k^2 M^2 + A_k.
$$

By summing up $k$ from 0 to n,

$$
\begin{aligned}
2\sum_{k=0}^{n} \gamma_k \mathbb{E}[g(\theta_k; L_k) - g(\theta_*^{L_k}; L_k)] &\leq -\mathbb{E}[\|\theta_{n+1} - \theta_*^{L_{n+1}}\|^2 - \|\theta_0 - \theta_*^0\|^2] + M^2 \sum_{k=0}^{n} \gamma_k^2 + \sum_{k=0}^{n} A_k \\
&\leq D^2 + M^2 \sum_{k=0}^{n} \gamma_k^2 + \sum_{k=0}^{n} A_k,
\end{aligned}
\tag{27}
$$

where $D = \|\theta_0 - \theta_*^0\|$. Dividing $2\sum_{k=0}^{n} \gamma_k$ on both sides, we have

$$
\frac{1}{\sum_{k=0}^{n} \gamma_k} \sum_{k=0}^{n} \gamma_k \mathbb{E}[g(\theta_k; L_k) - g(\theta_*^{L_k}; L_k)] \leq \frac{D^2 + M^2 \sum_{k=0}^{n} \gamma_k^2 + \sum_{k=0}^{n} A_k}{2\sum_{k=0}^{n} \gamma_k} \leq \frac{D^2 + M^2 \sum_{k=0}^{n} \gamma_k^2 + A}{2\sum_{k=0}^{n} \gamma_k}.
$$

According to the convexity of $g(\cdot; \cdot)$ and Jensen's inequality Jensen (1906),

$$
\frac{1}{\sum_{k=0}^{n} \gamma_k} \sum_{k=0}^{n} \gamma_k \mathbb{E}[g(\theta_k; L_k)] \leq \mathbb{E}[g(\bar{\theta}_n; \bar{L}_n)], \tag{28}
$$

where $\bar{\theta}_n = \frac{\sum_{k=0}^{n} \gamma_k \theta_k}{\sum_{k=0}^{n} \gamma_k}$ and $\bar{L}_n = \frac{\sum_{k=0}^{n} \gamma_k L_k}{\sum_{k=0}^{n} \gamma_k}$.

Then we have

$$
\mathbb{E}[g(\bar{\theta}_n; \bar{L}_n)] - \frac{\sum_{k=0}^{n} \gamma_k g(\theta_*^{L_k}; L_k)}{\sum_{k=0}^{n} \gamma_k} \leq \frac{D^2 + M^2 \sum_{k=0}^{n} \gamma_k^2 + A}{2\sum_{k=0}^{n} \gamma_k}, \tag{29}
$$

We choose $\gamma_k$ such that $\sum_{k=0}^{n} \gamma_k = \infty$ and $\sum_{k=0}^{n} \gamma_k^2 < \infty$, for example, $\gamma_k = \frac{1}{k}$. Taking $n$ to infinite, we have

$$
0 \leq \lim_{n \to \infty} \mathbb{E}[g(\bar{\theta}_n, \bar{t}_n)] - \lim_{n \to \infty} \frac{\sum_{k=0}^{n} \gamma_k g(\theta_*^{L_k}, L_k)}{\sum_{k=0}^{n} \gamma_k} \leq 0. \tag{30}
$$

Because $L_k \nearrow 0$, $g(\cdot, \cdot)$ is continuous and $\sum_{k=0}^{n} \gamma_k = \infty$, then

$$
\lim_{n \to \infty} \frac{\sum_{k=0}^{n} \gamma_k g(\theta_*^{L_k}, L_k)}{\sum_{k=0}^{n} \gamma_k} = g(\theta_*^0, 0), \tag{31}
$$

which implies that

$$
\lim_{n \to \infty} \mathbb{E}[g(\bar{\theta}_n, \bar{L}_n)] = g(\theta_*^0, 0). \tag{32}
$$

$\square$

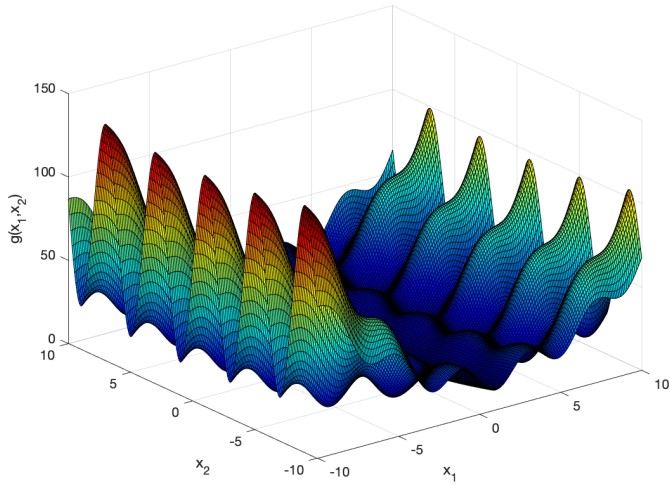

Figure 1: Two-dimensional highly non-convex function

## 4 NUMERICAL EXPERIMENTS

### 4.1 A NON-CONVEX EXAMPLE

We first use the following multi-dimensional highly non-convex example Chow et al. (2013) to illustrate the feasibility and efficiency of the HTA.

$$\min f(x) = (\pi/n)\left\{ k\sin^2(\pi y_1) + \sum_{i=1}^{n-1}(y_i - A)^2[1 + k\sin^2(\pi y_{i+1})] + (y_n - A)^2 \right\}, \quad (33)$$

where

$$x = (x_1, \cdots, x_n) \in \mathbb{R}^n, \quad y_i = 1 + (x_i - 1)/4, \quad i = 1, \cdots, n \text{ and } k = 10, \quad A = 1. \quad (34)$$

The function has roughly $5^n$ local minima in the region $\{|x_i| < 10\}$ and a unique global minimum located at $x_i = 1, \quad i = 1, \cdots, n$. (See Figure 4.1 for an illustration of $n = 2$.)

The idea of the homotopy setup of this problem is splitting the $n$ dimensional optimization problem into two $n/2$ dimensional problems which can be optimized separately and are tracked to an optimal of solution of $f(x)$ with respect to the homotopy parameter $L$. Then the objective function of the homotopy method is written as follows:

$$F(x, y; L) = f(x) + L\|(x_1, \cdots, x_{\frac{n}{2}}) - (x_1^0, \cdots, x_{\frac{n}{2}}^0)\|^2 + g(y) + L\|(y_{\frac{n}{2}+1}, \cdots, y_n) - (y_{\frac{n}{2}+1}^0, \cdots, y_n^0)\|^2$$

$$+ \frac{1}{L}\|x - y\|^2, \quad (35)$$

where

$$x = (x_1, \cdots, x_n) \in \mathbb{R}^n, \quad y = (y_1, \cdots, y_n) \in \mathbb{R}^n, \quad (36)$$

$(x_1^0, \cdots, x_{\frac{n}{2}}^0) \in \mathbb{R}^{\frac{n}{2}}$, $(y_{\frac{n}{2}+1}^0, \cdots, y_n^0) \in \mathbb{R}^{\frac{n}{2}}$ are fixed and chosen based on the initial guess. When $L$ is large, $F$ is equivalent to optimize

$$\min_{x_{\frac{n}{2}+1},\cdots,x_n} g(x_1^0, \cdots, x_{\frac{n}{2}}^0, x_{\frac{n}{2}+1}, \cdots, x_n) + \min_{y_1,\cdots,y_{\frac{n}{2}}} g(y_1, \cdots, y_{\frac{n}{2}}, y_{\frac{n}{2}+1}^0, \cdots, y_n^0), \quad (37)$$

which can be solved separately. When $L$ becomes small, the last term of (35) forces $x$ and $y$ to be equal. Then $F(x, y; L)$ is equivalent to optimizing $f(x)$ which is the original n-dimensional optimization problem. Therefore $F(x, y; L)$ connects two optimization problems by gradually decreasing penalty factor $L$ and builds a continuous path between the decoupled systems and the original optimization function $f(x)$.

| Dimensions (n) | Initial Values | Quasi-newton with the homotopy setup | Quasi-newton |
|---|---|---|---|
| 2 | $[0, 1, \cdots, 5]^2$ ($6^2$ points) | 36 | 24 |
| 4 | $[0, 1, \cdots, 5]^4$ ($6^4$ points) | 1296 | 864 |
| 6 | $[3, 4, 5, 6]^6$ ($4^6$ points) | 4096 | 991 |

Table 1: Finding global minimums by two algorithms

| Base Model Name | Original Error Rate | Error Rate with HTA | Rate of Imrpovement(ROIs) |
|---|---|---|---|
| VGG11 | 7.83% | 7.02% | 10.34% |
| VGG13 | 5.82% | 5.14% | 11.68% |
| VGG16 | 6.14% | 5.71% | 7.00% |
| VGG19 | 6.35% | 5.88% | 7.40% |

Table 2: The results of HTA and traditional method of VGG models on CIFAR-10

We compared the traditional optimization method (the quasi-Newton method) and the quasi-Newton with the homotopy setup. For $n = 4$, we chose the same initial value $[5, 4, 5, 5]$ and set $L$ from 0.9 to 0.1 for the quasi-newton with the homotopy setup. It takes 8 steps and 14 steps for the quasi-newton method and the quasi-newton with the homotopy setup to converge, respectively. However, the quasi-newton method does not find the global minimum while the HTA does. We also compared two algorithms with different initial values and count the times when converging to global minimums. Table 1 shows the results of $n = 2$, $n = 4$, $n = 6$ and $n = 8$.

## 4.2 APPLICATIONS ON VGG

Secondly, we applied the HTA to both CNN and fully connected neural networks on the CIFAR-10 dataset (`https://www.cs.toronto.edu/~kriz/cifar.html`) with the training and validation datasets from the CIFAR-10 website. In terms of the pre-processing step: for training dataset, we used random horizontal flip, random crop and normalization; for the validation dataset, we just used the same normalization. We implemented the HTA with stochastic gradient descent (SGD) method on VGG11, VGG13, VGG16 and VGG19 as our base models with batch normalization Simonyan & Zisserman (2014). In order to compare the SGD with the HTA method, we set all the regular parameters (e.g. learning rate, batch size, momentum, epochs, weight decay and etc) to be exactly the same, varied the homotopy parameter $L$ from $L = 0.01$ to $L = 0.005$ with a stepsize $\Delta L = 0.001$. For each $L$, we run 4 epochs with the SGD method. The code for this method can be found at `https://github.com/Bill-research/homotopy` which has been run on the DGX-1 (p100 version) with 8 p100 GPUs and Tesla structures. Figure 2 shows the comparison of validation loss between the HTA with fully connected neural networks and traditional method on the VGG13 model. The HTA has a less error rate (5.14%) than the traditional method (6.35%) by improving 11.68%. All the results for different models are shown in Table 2. It is clearly seen that accuracies of HTA for different models are better than that of the traditional method. For example, vgg11 with HTA gets 7.02% error rate while traditional method only gets 7.83%; The vgg16 with HTA has an error rate of 5.88% while traditional method only has 6.35%. We run the numerical experiments for 5 times for each model and the results are shown in Table 3. We also applied HTA to the only CNN part of two VGG models: vgg11 and vgg13. Figure 3 shows the validation accuracies of two models which has been improved by the HTA (91.61% (vgg11) and 93.09% (vgg13)) comparing with the traditional method (91.01% (vgg11) and 92.29% (vgg13)).

| Base Model Name | mean | stddev | max |
|---|---|---|---|
| VGG11 | 91.53% | 0.09% | 91.66% |
| VGG13 | 92.76% | 0.22% | 93.09% |

Table 3: The validation accuracies of HTA method of VGG models on CIFAR-10

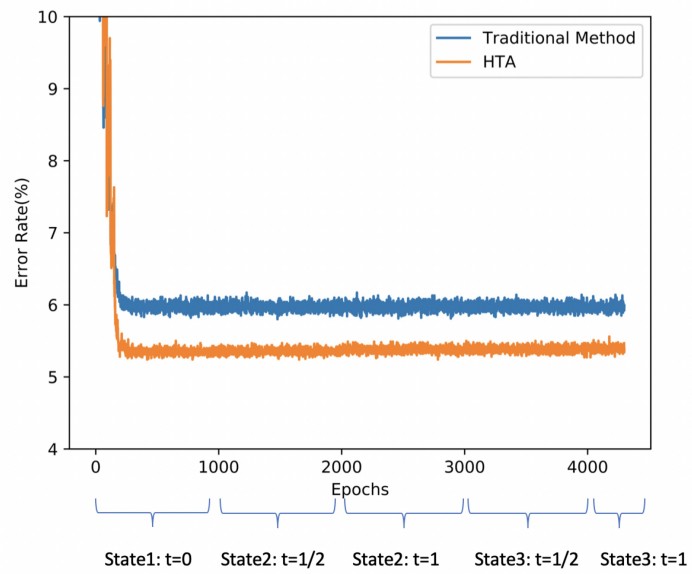

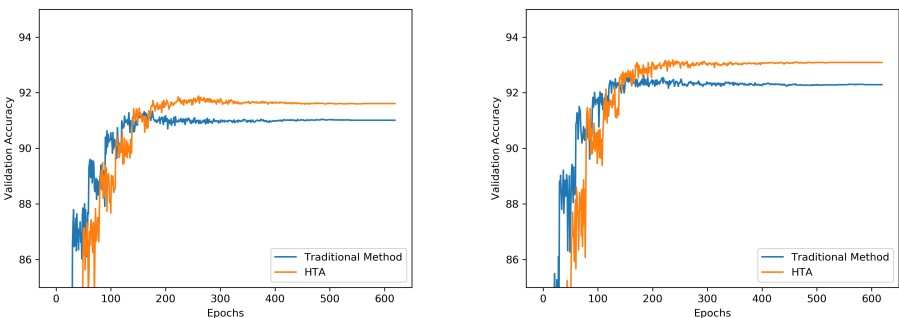

Figure 2: Comparison of error rate of VGG13 between HTA and the traditional method.

Figure 3: The performance of the HTA on the CNN part of vgg11 (left) and vgg13 (right) and comparisons with the traditional method

## 5 CONCLUSION

In this paper, we developed a homotopy training algorithm for solving the optimization problem arising from neural networks. This algorithm starts from decoupled low dimensional systems and gradually transforms to the coupled original system with complex structure. Then the complex neural networks can be trained by the HTA with a better accuracy. The convergence of the HTA for any given $L$ is proved for the non-convex optimization. Then existence of solution path $\theta(L)$ is demonstrated theoretically for the convex case although that it exists numerically in the non-convex case. Several numerical examples have used to demonstrate the efficiency and feasibility of HTA. The application of HTA to VGG models on CIFAR-10 provides a better accuracy than the traditional method. Moreover, the HTA method would provide a new way to couple with the dropout techniques to improve the training accuracy which we will explore in the future.

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
