# OpenReview forum: "AN EFFICIENT HOMOTOPY TRAINING ALGORITHM FOR NEURAL NETWORKS"
_ICLR.cc/2020/Conference — Reject_

### Official Review · AnonReviewer1 · 2019-10-22
**Official Blind Review #1**

**Rating:** 3

**Review:**

In this paper, the authors propose the Homotopy Training Algorithm (HTA) for neural network optimization problems. They claim that HTA starts with several simplified problems and tracks the solution to the original problem via a continuous homotopy path. They give the theoretical analysis and conduct experiments on the synthetic data and the CIFAR-10 dataset.
My major concerns are as follows.
1. The authors may want to give more detailed explanations of HTA. For example, they may want to give the pseudocode for HTA and explain its advantages compared to other optimization methods.
2. The theoretical analysis is trivial. The proof of Theorem 3.1 is to verify Assumptions 4.1 and 4.3 in [1]. Moreover, the proof of Theorem 3.2 is similar to the analysis for the convergence of SGD for convex problems in [2].
3. The experiments do not show the efficiency of HTA, as the original quasi-newton method is faster than the quasi-newton method with the homotopy setup.
4. The authors make a mistake in the proof of Theorem 3.1. The claim that “{\theta_k} is contained in an open set which is bounded. Since that g is continuous, g is bounded.” is incorrect. We can find a counterexample g(x) = \frac{1}{x}, x\in (0,1).

[1] L. Bottou, F. Curtis, and J. Nocedal. Optimization methods for large-scale machine learning. SIAM Review, 60(2):223–311, 2018.
[2] A. Nemirovski, A. Juditsky, G. Lan, and A. Shapiro. Robust stochastic approximation approach to stochastic programming. SIAM Journal on Optimization, 19(4):1574–1609, 2009.

**Experience Assessment:**

I have published in this field for several years.

**Review Assessment: Checking Correctness Of Derivations And Theory:**

I carefully checked the derivations and theory.

**Review Assessment: Checking Correctness Of Experiments:**

I assessed the sensibility of the experiments.

**Review Assessment: Thoroughness In Paper Reading:**

I read the paper at least twice and used my best judgement in assessing the paper.

---

### Official Review · AnonReviewer3 · 2019-10-27
**Official Blind Review #3**

**Rating:** 3

**Review:**

The work proposes to learn neural networks using homotopy-based continuation method. The method divides the parameter space into two groups (extendable to multiple groups) and introduces a homotopy function which includes the original optimization problem as an extreme case.  By varying the homotopy parameter, one can construct a continuous path from a supposedly easier to solve optimization problem to the problem of interest. The authors prove convergence in the non-convex case, the existence of solution path in the convex case and demonstrate the effectiveness of the proposed method on synthetic and real datasets.

While the idea itself is rather intriguing and seems promising, the current presentation and experimentation does not meet the acceptance threshold.  The writing of the draft needs a lot of improvement, in particular the notations the authors used are not consistent throughout the paper, which is very confusing.

The synthetic example the authors used in section 4.1 are naturally decoupled among the different dimensions of the parameters, which is no surprise the proposed method would achieve 100% convergence as shown in table 1.

It seems the division of the parameter space would matter. One would imagine there exists certain division leading to much easier to solve subproblems. Do the authors have any insight or experiments comparing different division strategies?

Here's a very closely-related work that should be cited and discussed:
Wang, Xin. "An efficient training algorithm for multilayer neural networks by homotopy continuation method." Proceedings of 1994 IEEE International Conference on Neural Networks (ICNN'94). Vol. 1. IEEE, 1994.


Typos:
1) Equation following remark in page 2, should H() be replaced by G() or \nabla H()?
2) After equation (7), should G():= \nabla H instead of F?


**Experience Assessment:**

I do not know much about this area.

**Review Assessment: Checking Correctness Of Derivations And Theory:**

I did not assess the derivations or theory.

**Review Assessment: Checking Correctness Of Experiments:**

I carefully checked the experiments.

**Review Assessment: Thoroughness In Paper Reading:**

I read the paper at least twice and used my best judgement in assessing the paper.

---

### Official Review · AnonReviewer2 · 2019-11-03
**Official Blind Review #2**

**Rating:** 1

**Review:**

Summary

This paper proposes an algorithm to address the issue of nonlinear optimization
in high dimensions and applies it to convolutional neural networks (VGG models) on CIFAR 10.
They show 11% relative reduction in error for this particular task with this
particular network. In addition, they prove additional theoretical results on
the convergence of SGD using their method in the convex case as well as
convergence of SGD to a stationary point in the nonconvex case when the homotopy
parameter is fixed which is not done in practice.

Given an optimization problem, their method first solves multiple independent
lower-dimensional optimization problems each with a subset of the parameters and
then optimizes a new objective function controlled by a monotonically decreasing
parameter L that interpolates the original objective function and the
previously-solved lower dimensional problems. L can be seen as a regularization
parameter that is gradually decreased as we optimize the new optimization
function. When L = 0, we recover the original optimization problem.
The authors prove that (1) SGD with their procedure will find a stationary point
under the Robbins-Monro conditions for a fixed L and (2) SGD with their
procedure will converge for convex problems as L is decreased to 0.

Decision and reasoning

This paper should be rejected because (1) the proposed algorithm attempts to address
the original issue of high dimensional nonlinear optimization of neural networks but
violates the algorithm's assumption in practice, (2) the
empirical evaluations are lacking - having only evaluated their method on a toy
problem with up to only 6 dimensions and a relatively simple image classification task,
and (3) the assumption of fixing the homotopy parameter in the theorem on the
non-convex case directly violates the intention of the algorithm.

Regarding (1): The proposed procedure requires initializing L at a large value
and reducing L towards 0 in order to recover the original optimization problem.
However, in practice for CIFAR 10, the authors initialize L to be 0.01 and
gradually reduces it to 0.005 which is hardly the original intent of the
algorithm. There is also no demonstration whether or not this gradual reduction in
L actually has an effect on the optimization of the new objective function. For
example, since the start and end values of L are similar, will we get similar
results if we simply fix L to be 0.005 or 0.01? The authors also show that their
method outperforms a quasi-newton method by combining the optimization with
their procedure on a non-convex example by Chow et al. 2013. However, this example
only goes up to n=6 dimensions, which is hardly comparable to the original
problem of high dimensional non-convex optimization that this paper sought to
address.

Regarding (2): The authors evaluated their procedure on CIFAR10, a relatively
simple image classification task that modern neural networks can solve easily
and is not representative of the types of nonlinear optimization problems
prevalent in deep learning. There's also an issue of using only VGG networks for
their evaluations while VGGs are typically eschewed in favor of ResNets today.
Given that the optimization is easier with residual connections, it may be the case that
their procedure does not significantly improve the accuracy of ResNets.

Regarding (3): By fixing L in Theorem 3.1, the authors essentially show that SGD
converges to a stationary point for their new objective function which can be
seen as a regularized version of the original objective function, which is not a
strong result. Furthermore, fixing L goes against the original procedure's
motivation of recovering the original optimization function as L decreases to 0.

Additional comments and questions

There are passages that are difficult to understand because not enough context
is given. For example in the "remark" passage, it is not clear where the
"necessary condition" comes from. In addition it seems like it doesn't even
type-check since the first term is 2n dimensional while the second term is 4n
dimensional.

There are also many errors in the writing that hinder the presentation. A subset of
them includes:
- "nerual netowrks on roboticsKonda et al." -> "neural networks on robotics Konda et al."
- "based on homotopy continuation method" -> "based on the homotopy continuation methods"
- "random chosen point" -> "randomly chosen point"
- "we choose \tilde{\theta} = 0 in the dropout" -> reword
- Fourth term in Equation 3 should be \theta_2 - \tilde{\theta_2}
- "By gradually increasing parameter L" -> "By gradually decreasing parameter L"
- "where \xi is a random variable due to random algorithms" -> reword and possibly say the randomness is from SGD
- After equation 6, should have b_i instead of \beta_i
- In equation 20, should be g(\theta_*^0) instead of g(\theta_*^1)
- In theorem 3.2 you never explained what \theta_*^{L_k} is
- "We compared the traditional optimization method (the quasi-Newton method)" -> which quasi-Newton method?
- Figures 2 and 3 label the x-axis with "epochs". However only 4 epochs were run, so I believe the x-axis should be "iterations""

Besides improving the quality of writing in the paper, I would strongly suggest that the
authors improve their empirical evaluation.  Possibilities include evaluating on
CIFAR 100 or ImageNet, using a wider variety of networks including ResNets,
evaluating on tasks other than image classification.

**Experience Assessment:**

I have read many papers in this area.

**Review Assessment: Checking Correctness Of Derivations And Theory:**

I assessed the sensibility of the derivations and theory.

**Review Assessment: Checking Correctness Of Experiments:**

I carefully checked the experiments.

**Review Assessment: Thoroughness In Paper Reading:**

I read the paper thoroughly.

---

### Decision · Program_Chairs · 2019-12-19

**Decision:**

Reject

**Comment:**

The work proposes to learn neural networks using a homotopy-based continuation method. Reviewers found the idea interesting, but the manuscript poorly written, and lacking in experimental results. With no response from the authors, I recommend rejecting the paper.